# Vegetative cells may perform nitrogen fixation function under nitrogen deprivation in *Anabaena* sp. strain PCC 7120 based on genome-wide differential expression analysis

**Hongli He, Runyu Miao, Lilong Huang, Hongshan Jiang, Yunqing Cheng***

Jilin Provincial Key Laboratory of Plant Resource Science and Green Production, Jilin Normal University, Siping, Jilin Province, China

* chengyunqing1977@163.com

**Data Availability Statement:** All relevant data are within the manuscript and its Supporting information files.

## Abstract

Nitrogen assimilation is strictly regulated in cyanobacteria. In an inorganic nitrogen-deficient environment, some vegetative cells of the cyanobacterium *Anabaena* differentiate into heterocysts. We assessed the photosynthesis and nitrogen-fixing capacities of heterocysts and vegetative cells, respectively, at the transcriptome level. RNA extracted from nitrogen-replete vegetative cells (NVs), nitrogen-deprived vegetative cells (NDVs), and nitrogen-deprived heterocysts (NDHs) in *Anabaena* sp. strain PCC 7120 was evaluated by transcriptome sequencing. Paired comparisons of NVs vs. NDHs, NVs vs. NDVs, and NDVs vs. NDHs revealed 2,044 differentially expressed genes (DEGs). Kyoto Encyclopedia of Genes and Genomes enrichment analysis of the DEGs showed that carbon fixation in photosynthetic organisms and several nitrogen metabolism-related pathways were significantly enriched. Synthesis of *Gvp* (Gas vesicle synthesis protein gene) in NVs was blocked by nitrogen deprivation, which may cause *Anabaena* cells to sink and promote nitrogen fixation under anaerobic conditions; in contrast, heterocysts may perform photosynthesis under nitrogen deprivation conditions, whereas the nitrogen fixation capability of vegetative cells was promoted by nitrogen deprivation. Immunofluorescence analysis of nitrogenase iron protein suggested that the nitrogen fixation capability of vegetative cells was promoted by nitrogen deprivation. Our findings provide insight into the molecular mechanisms underlying nitrogen fixation and photosynthesis in vegetative cells and heterocysts at the transcriptome level. This study provides a foundation for further functional verification of heterocyst growth, differentiation, and water bloom control.

## Introduction

Cyanobacteria evolved approximately 3.5 billion years ago. As the first photosynthetic prokaryotes on earth, cyanobacteria can efficiently fix atmospheric nitrogen via a process catalyzed by nitrogenase [1]. Nitrogenase-catalyzed substrate reduction requires the association of

**Funding:** This study was supported by grants from the National Natural Science Foundation of China (No. 31670681; 31770723).

**Competing interests:** The authors declare that they have no conflict of interest.

an iron (Fe) protein and molybdenum-iron (MoFe) protein, with electron transfer occurring from the Fe protein to the MoFe protein. However, nitrogenase responsible for the reduction of $N_2$ is extremely sensitive to $O_2$ [2–5]. How to avoid oxygen released during photosynthesis by nitrogenase has attracted much attention. Cyanobacteria need to segregate the timing of nitrogen fixation from oxygenic photosynthesis by restricting nitrogen fixation to the dark period of diel cycles or spatially separate these processes by restricting nitrogen fixation to heterocysts [6, 7]. Under nitrogen-deprivation conditions, heterocysts differentiate from vegetative cells only in the filaments of cyanobacteria [8]. It is well-known that vegetative cells perform photosynthetic functions, whereas heterocysts perform nitrogen fixation functions in cyanobacteria. However, whether both cell types perform photosynthesis and nitrogen fixation simultaneously remains unclear.

All aerobic nitrogen fixation occurs in heterocysts in a semiregular pattern, and nitrogen fixed in heterocysts is transported to vegetative cells in the filament; vegetative cells supply carbon and reductants to heterocysts [6, 9]. The heterocyst is the site of dinitrogen fixation and provides oxygen-sensitive nitrogenase with a low-oxygen environment [10]. In general, heterocysts lack photosystem II activity and ribulose bisphosphate carboxylase, and they cannot photoreduce $CO_2$ via the reductive pentose phosphate pathway to provide carbon skeletons for assimilation of fixed nitrogen [6, 11–13]. The heterocyst develops a special glycolipid layer that serves as a gas diffusion barrier, and the heterocyst glycolipid layer can be modified in response to the external O2 concentration [10, 14]. However, other studies suggested that heterocysts are not the only cells capable of nitrogen fixation in heterocystous cyanobacteria and that vegetative cells can fix molecular nitrogen using another nitrogenase encoded by a homologous gene cluster named as nif2 [15]. Mutants of heterocystous cyanobacteria (*het⁻*) that fail to produce heterocysts retain their nitrogen fixation ability when incubated under micro-oxic or anoxic conditions [16]. Thus, studies are needed to evaluate the nitrogen fixation ability of vegetative cells and photosynthetic ability of heterocysts in cyanobacteria.

Recent genome-wide studies have focused on cell-specific metabolism in cyanobacteria by comparing gene transcript levels under different growth conditions or stage-specific gene expression signatures in response to nitrogen step-down. RNA sequencing was used to study transcript expression levels in filaments at different periods of nitrogen step-down using a mixture of heterocysts and vegetative cells [17–19]. Using samples collected during the differentiation in three developmental states, a DNA microarray containing 6,893 gene fragments were was used to identify differentially expressed genes (DEGs) regulating the differentiation of heterocysts, akinetes, and hormogonia in the cyanobacterium *Nostoc punctiforme* [20]. Similarly, seven genes encoding transcriptional regulators that respond to nitrogen deprivation in *Anabaena* sp. strain PCC 7120 were identified by oligonucleotide microarray technology, and the results for an *nrrA* deletion mutant suggested that *nrrA* facilitates heterocyst development [21]. However, vegetative cells in the mycelium samples were not separated from heterocysts in the filamentous cyanobacteria. Therefore, the function of vegetative cells and heterocysts may have been biased in sequencing and bioinformatics analysis. Compared with vegetative cells, heterocysts cells of *Anabaena* have thicker walls and better resist adverse external environments, providing a theoretical basis for the separation and purification of the two types of cells. In general, two strategies are used to isolate and purify heterocysts from the mycelium: sonication treatment [22] or lysozyme treatment [23]. Vegetative cells easily crack during these treatments, resulting in the isolation of heterocysts without vegetative cells. Heterocysts can be isolated from mycelium containing both vegetative cells and heterocysts of *Anabaena* sp. strain 7120 with lysozyme for RNA-Seq analysis (Fig 1). We evaluated important candidate genes involved in nitrogen fixation and photosynthesis in vegetative cells and heterocysts from *Anabaena* sp. strain PCC 7120 at the

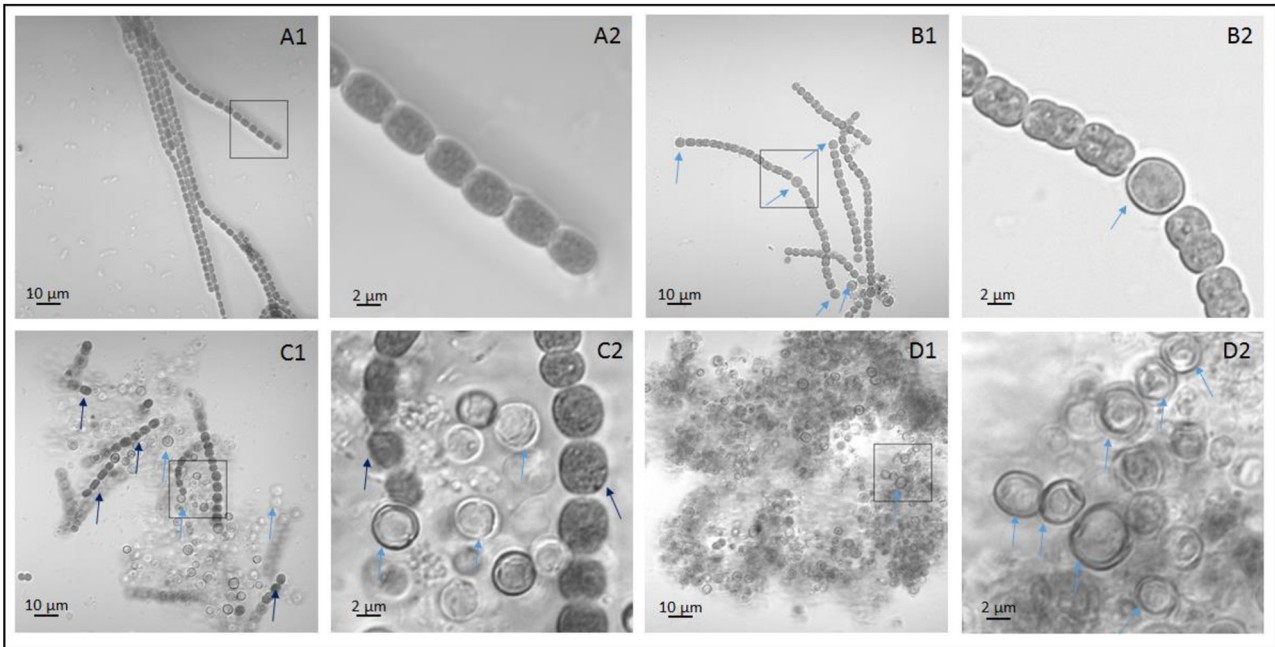

**Fig 1. Enrichment and purification heterocysts in *Anabaena* sp. strain PCC 7120.** (A1 and A2) Filaments only consisting of vegetative cells under nitrogen-replete condition. A2 is an enlarged view of black square in A1. (B1 and B2) Filaments consisting of vegetative cells and heterocysts under nitrogen deprivation condition. B2 is an enlarged view of black square in B1. (C1 and C2) The filaments were treated with lysozyme followed by centrifuge. The picture shows the mixture of heterocysts and residual vegetative cells in the sediment. The supernatant was used to extract mRNA from vegetative cells. C2 is an enlarged view of black square in C1. (D1 and D2) Pure heterocysts for RNA extraction. D2 is an enlarged view of black square in D1. Black and blue arrows show vegetative cells and heterocysts respectively.

transcriptome level and provide insight into the molecular mechanisms of nitrogen fixation and photosynthesis in both cell types.

## Results

### Illumina sequencing and clean read mapping

To evaluate entire gene expression profiles in nitrogen-deprived vegetative cells (NDVs), nitrogen-deprived heterocysts (NDHs), and nitrogen-replete vegetative cells (NVs), nine digital gene expression (DGE) profiling libraries of *Anabaena* sp. strain PCC 7120 were sequenced using the Illumina HiSeq 4000 platform, which generated approximately 26 million clean reads for each library (S1 Table), accounting for approximately 92% of the total obtained reads. In total, 243,139,646 clean reads were acquired, which were 36,714,086,546 bp in length; the average length of each clean read was 151 bp (S1 Table). For all reads in the nine libraries, statistical analysis of the distribution results of RNA-Seq mapped events showed that approximately 7%, 93%, 78%, and 1% mapped reads were queried against the intergenic regions, genes, mRNA, and rRNA, respectively, of the reference genome, suggesting that most reads were from gene sequences (S2 Table). For all nine libraries, clean read assembly generated 5,842 genes with a mean length of 988 bp.

### Global gene expression in NDVs, NDHs, and NVs

We found that 5,808, 5,812, and 4,792 genes were expressed in NVs, NDVs, and NDHs, respectively. The number of expressed genes in NDHs was smaller than that in NVs and NDVs. These results suggest that more genes are necessary for the growth of vegetative cells

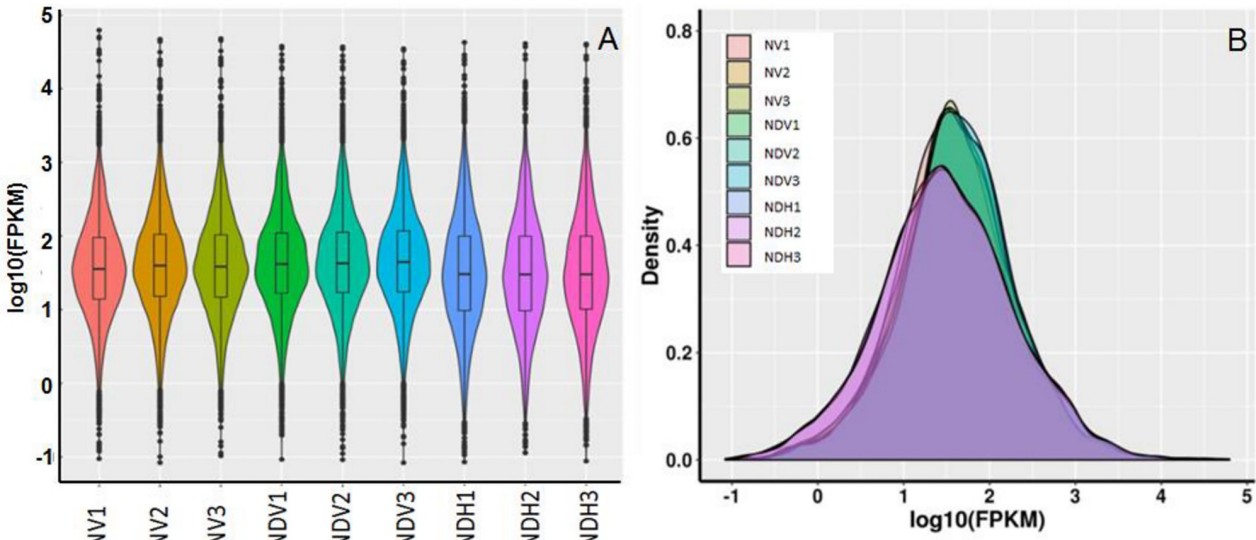

**Fig 2. General distribution of gene expression in *Anabaena* sp. strain PCC 7120.** (A) The FPKM boxplot distribution is shown with a box plot. (B) FPKM density distribution for all transcripts. Note: FPKM, fragments per kilobase of transcript per million mapped reads. 'NDV' and 'NDH' refers to vegetative cells and heterocysts cultured respectively under nitrogen deprivation condition, and 'NV' refers to vegetative cells cultured under nitrogen-replete condition.

than for the growth of heterocysts. Furthermore, the number of expressed genes in NVs and NDVs was similar. Fragments per kilobase of transcript per million mapped reads (FPKM) values were used to calculate gene expression levels; the results suggested that the average FPKM values of NDHs were lower than those of NVs and NDVs (Fig 2A). Consistent with these findings, the densities of NDH genes with medium and low FPKM values were lower and higher, respectively, than those of NV and NDV samples, suggesting relatively low expression of genes in NDHs (Fig 2B).

## Identification of DEGs in three paired comparisons

In total, 2,044 DEGs were identified in NV vs. NDH, NV vs. NDV, and NDV vs. NDH comparisons (Fig 3A). Among these, 251, 420, and 149 DEGs were uniquely expressed in NDV vs. NDH, NV vs. NDH, and NV vs. NDV samples, respectively, and 116 DEGs were commonly expressed in these comparisons (Fig 3B).

## Kyoto Encyclopedia of Genes and Genomes (KEGG) and Gene Ontology (GO) enrichment analysis of DEGs

KEGG significant enrichment analysis of DEGs can reveal metabolic pathway and signaling pathway information. KEGG functional enrichment analysis of all DEGs was performed to determine the pathways involved in regulating the responses of vegetative cells and heterocysts to nitrogen deprivation conditions. Fourteen significantly enriched pathways were identified, and 7, 3, and 4 KEGG pathways were enriched in NV vs. NDH, NV vs. NDV, and NDV vs. NDH comparisons, respectively (Table 1). In the NDV vs. NDH comparison, five pathways were related to metabolism, including those related to carbohydrates, nucleotides, and biosynthesis of other secondary metabolites. In the NV vs. NDV comparison, two pathways were involved in energy and amino acid metabolism and one pathway was involved in replication and repair. In the NDV vs. NDH comparison, three pathways were related to the metabolism

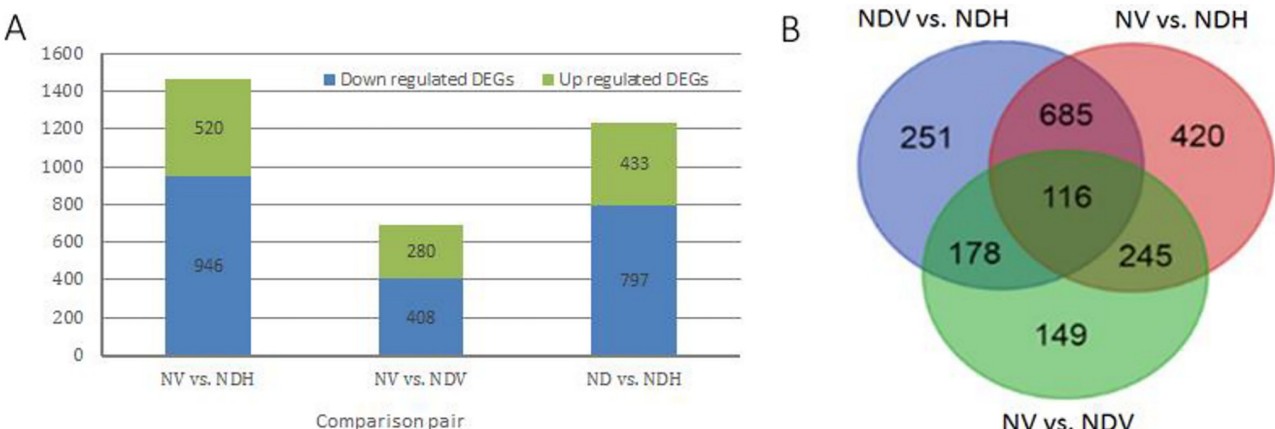

**Fig 3. Differentially expressed genes (DEGs) in three paired comparison of *Anabaena* sp. strain PCC 7120.** (A) DEGs amount in each paired comparison; (B) Venn diagram of all identified DEGs. In three paired comparisons, X vs. Y refers to Y/X; 'NDV' and 'NDH' refers to vegetative cells and heterocysts cultured respectively under nitrogen deprivation condition, and 'NV' refers to vegetative cells cultured under nitrogen-replete condition.

of cofactors and vitamins, terpenoids and polyketides, and nucleotides. These results suggest that most DEGs were involved in metabolism, carbon fixation, and the synthesis of important nitrogenous biomacromolecules including, but not limited to, arginine, DNA, and purine, providing important insight into screening of candidate genes involved in regulating vegetative cells and heterocysts in response to nitrogen deprivation conditions.

To determine the main biological functions of the identified DEGs, GO functional enrichment analysis of the DEGs was performed. All 2,044 DEGs were assigned to three GO categories, including biological processes, cellular components, and molecular functions (Fig 4). Most of these DEGs were involved in the following functions: cofactor transferase activity, cofactor binding, substrate-specific transporter activity, catalytic activity, acting on a protein, and transition metal ion binding. Moreover, most DEGs contributed to cellular and metabolic

**Table 1. Significantly enriched KEGG pathway in three comparison pairs.**

| Comparison pair | Pathway ID | Pathway | TGN | *P*-value |
|---|---|---|---|---|
| NDV vs. NDH | ko00520 | Amino sugar and nucleotide sugar metabolism | 24 | 0.025 |
| | ko00230 | Purine metabolism | 61 | 0.028 |
| | ko02040 | Flagellar assembly | 37 | 0.034 |
| | ko00525 | Acarbose and validamycin biosynthesis | 2 | 0.035 |
| | ko00521 | Streptomycin biosynthesis | 11 | 0.040 |
| | ko05133 | Pertussis | 11 | 0.040 |
| | ko00562 | Inositol phosphate metabolism | 5 | 0.049 |
| NV vs. NDV | ko00710 | Carbon fixation in photosynthetic organisms | 17 | 0.024 |
| | ko00220 | Arginine biosynthesis | 30 | 0.028 |
| | ko03030 | DNA replication | 20 | 0.046 |
| NV vs. NDH | ko00790 | Folate biosynthesis | 29 | 0.008 |
| | ko01051 | Biosynthesis of ansamycins | 3 | 0.012 |
| | ko02024 | Quorum sensing | 59 | 0.020 |
| | ko00230 | Purine metabolism | 61 | 0.029 |

KEGG: Kyoto Encyclopedia of Genes and Genomes; TGN: Total DEGs number. In three paired comparisons, X vs. Y refers to Y/X.

processes, including those pertaining to organic cyclic compounds, heterocyclic compounds, small molecules, and phosphorus. Most corresponding proteins were located in the cell membrane including the intrinsic and integral parts of the membrane.

## Cluster analysis of DEGs

To explore the relationship between the expression pattern of these DEGs and their biological functions, we performed cluster analysis of all identified DEGs. Hierarchical clustering analysis of the 2,044 DEGs in NV vs. NDH, NV vs. NDV, and NDV vs. NDH paired comparisons indicated that they could be classified into nine clusters, and DEGs belonging to a given cluster showed a similar expression pattern (Figs 4B and 5A). Among all DEGs, a few highly expressed DEGs were found in clusters 1, 4, 5, and 6. Many DEGs in cluster 1 were highly expressed in NDVs and NDHs, whereas their expression in NVs was relatively lower, and most highly expressed DEGs ($|\log_2$fold-change$| > 3$) were involved in nitrogen metabolism. The expression levels of highly expressed DEGs ($|\log_2$fold-change$| > 3$) in clusters 4, 5, and 6 differed in the NDH and NDV samples; some were involved in gas vesicle protein biosynthesis. Among the highly expressed DEGs belonging to clusters 6, 7, and 9, genes encoding phosphate ABC transporter permease, transposase, and cation efflux system proteins were strongly downregulated in the NDH sample.

## DEG validation by qRT-PCR and RT-PCR

To validate the DEGs identified in the NV vs. NDH, NV vs. NDV, and NDV vs. NDH paired comparisons by RNA-Seq technology, four DEGs that may be involved in nitrogen fixation and photosystems were selected for validation by qRT-PCR analysis, and two DEGs that may be involved in nitrogen fixation were selected for validation by RT-PCR analysis. In general, the $\log_2$ (fold-change) values obtained using the Illumina sequencing platform were consistent with those obtained by qRT-PCR analysis (Fig 6). Results of RT-PCR tests were shown in S1 Fig. These results indicate that our mRNA sequencing results were reliable.

## Immunofluorescence analysis of nitrogenase iron protein (NifH)

4',6-Diamidino-2-phenylindole (DAPI) and fluorescein 5-isothiocyanate (FITC) were used to visualize NifH expression in the filaments of *Anabaena* sp. strain PCC 7120. DAPI is a

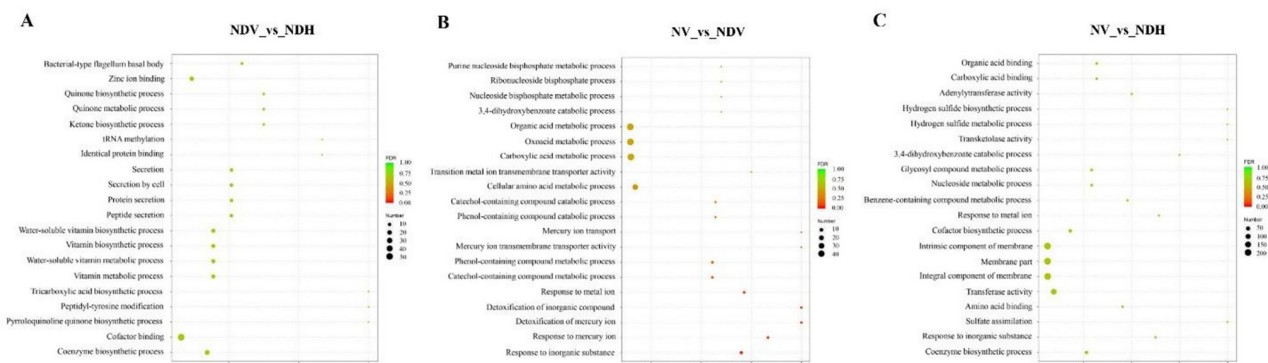

**Fig 4. The 20 most enriched GO terms of different expression genes in *Anabaena* sp. strain PCC 7120.** (A) NDV vs. NDH; (B) NV vs. NDV; (C) NV vs. NDH. "Rich factor" means that the ratio of the DEGs number and the number of genes have been annotated in this term. The greater of the Rich factor, the greater the degree of enrichment. The different color represents different false discovery rate (FDR) values. 'NDV' and 'NDH' refers to vegetative cells and heterocysts cultured respectively under nitrogen deprivation condition, and 'NV' refers to vegetative cells cultured under nitrogen-replete condition. In three paired comparisons, X vs. Y refers to Y/X.

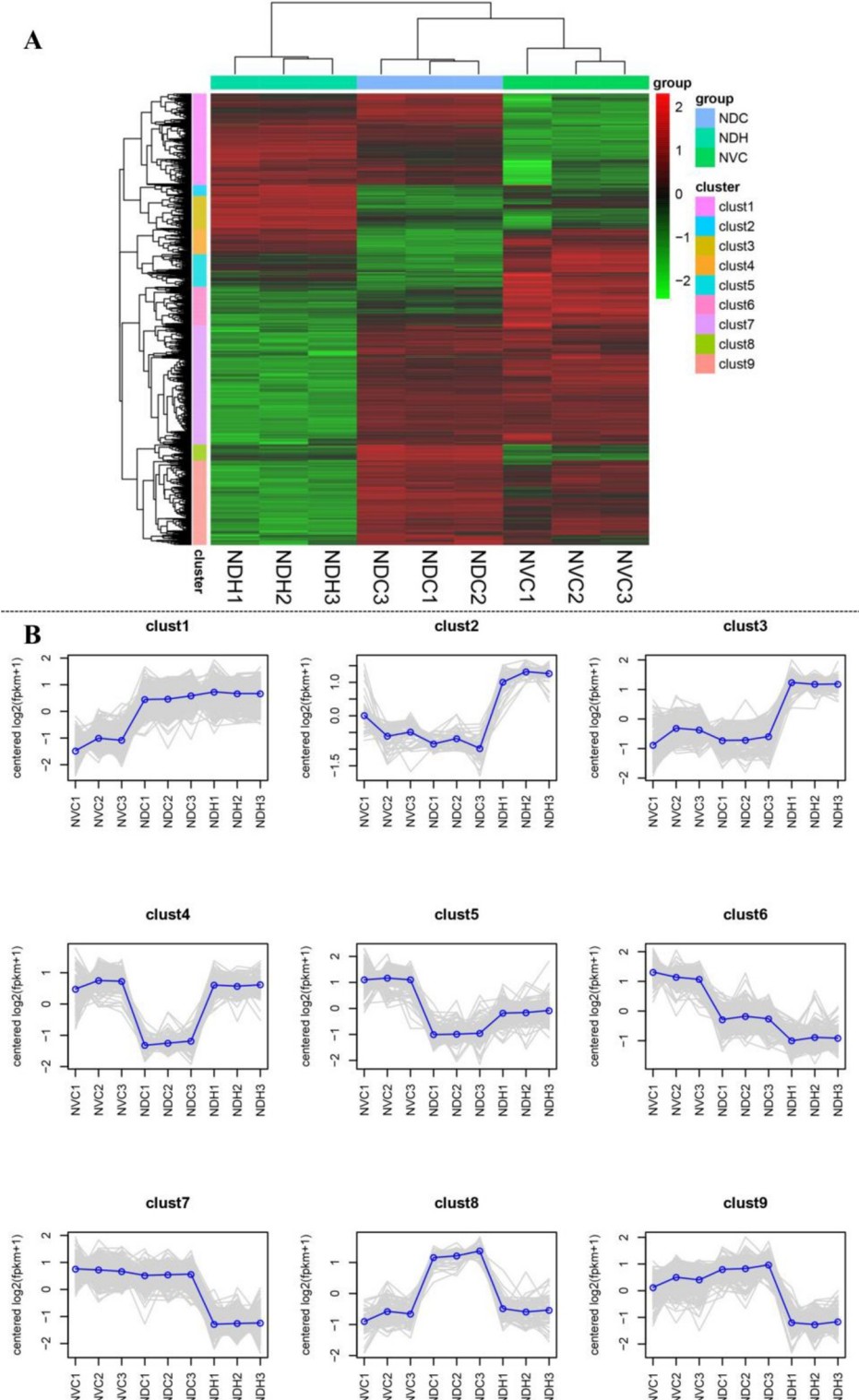

**Fig 5. Differential expressed genes cluster analysis of vegetative cells and heterocysts from *Anabaena* sp. strain PCC 7120.** (A) Heat map of the hierarchical cluster analysis of gene expression in NDH, NDV and NV. Each row represents a single gene. Each column represents a single sample. Green squares indicate transcript levels below the mean; black squares, transcript levels equal to the mean; red squares, transcript levels greater than the mean; gray squares, technically inadequate or missing data. Colored bars leftmost adjacent to groups in which the cluster of genes is expressed. (B) Trend analysis of different clusters. The gray line in the figure shows the expression pattern of

different DEGs in each cluster. The blue line represents the average expression level of all DEGs in the cluster in different samples. 'NDV' and 'NDH' refers to vegetative cells and heterocysts cultured respectively under nitrogen deprivation condition, and 'NV' refers to vegetative cells cultured under nitrogen-replete condition.

fluorescent dye that can tightly combine with DNA. Our second antibody of NifH was labeled with FITC. DAPI and FITC emitted blue and green fluorescence after excitation respectively. Under nitrogen-replete condition, filaments consisting of only vegetative cells (Fig 7A1–7A4), and weak green fluorescence of vegetative cells were detected (Fig 7A3). Under nitrogen deprivation conditions, in filaments consisting of vegetative cells and heterocysts, there was one heterocyst in every approximately 10 cells (Fig 7B1–7B4). Compared with vegetative cells cultured under nitrogen-replete conditions, the green fluorescence intensity of vegetative cells cultured under nitrogen deprivation conditions was clearly stronger (Fig 7A3 and 7B3), indicating that nitrogen deprivation induced the expression of NifH. Under nitrogen deprivation conditions, the green fluorescence of vegetative cells was not lower than that of heterocysts. Thus, there was almost no difference in the NifH expression level between vegetative cells and heterocysts under nitrogen deprivation conditions.

## Discussion

### Nitrogen deprivation induces heterocyst differentiation and inhibits gas vesicle development in vegetative cells

Gas vesicles are gas-filled prokaryotic organelles that function as flotation devices [24–26]. A range of bacteria and archaea produces intracellular gas-filled proteinaceous structures to

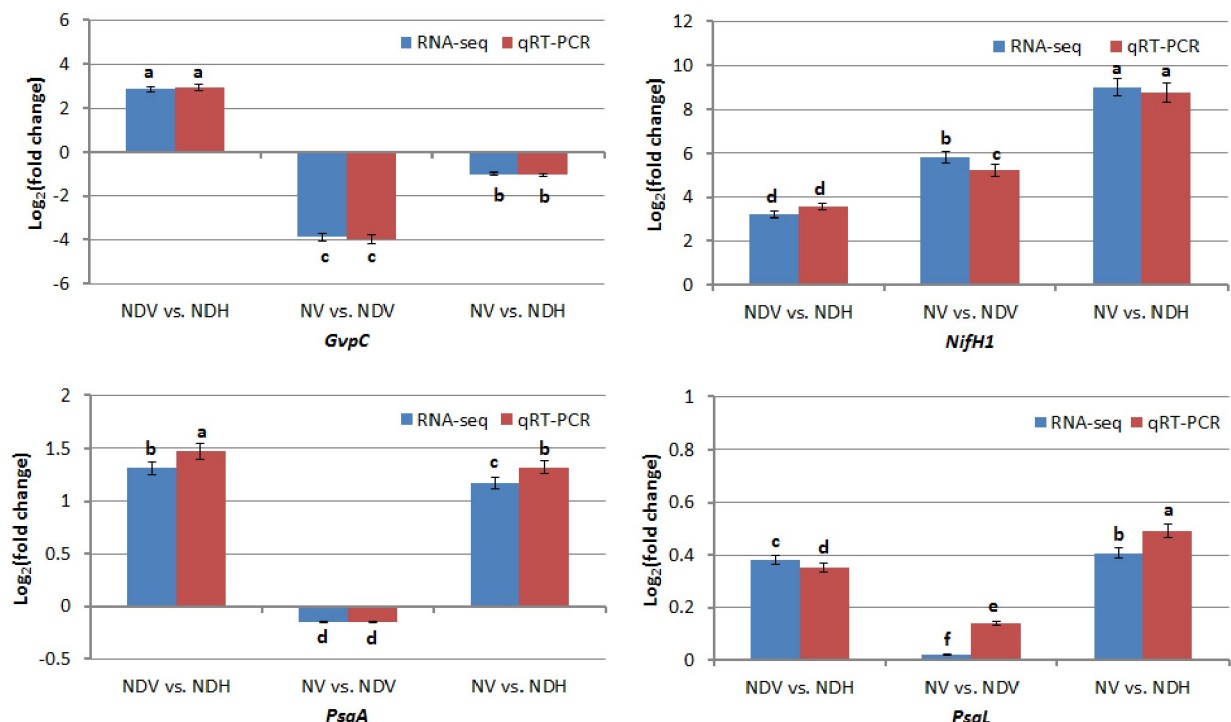

**Fig 6. Validation of DEGs by qRT-PCR analysis in *Anabaena* sp. strain PCC 7120.** The relative expression levels of four chosen DEGs were obtained by RNA-seq and qRT-PCR. Bars represent mean ± standard deviation (n = 3). Different low case letter above each column indicated significant difference at $P$ = 0.05. 'NDV' and 'NDH' refers to vegetative cells and heterocysts cultured respectively under nitrogen deprivation condition, and 'NV' refers to vegetative cells cultured under nitrogen-replete condition. In three paired comparisons, X vs. Y refers to Y/X.

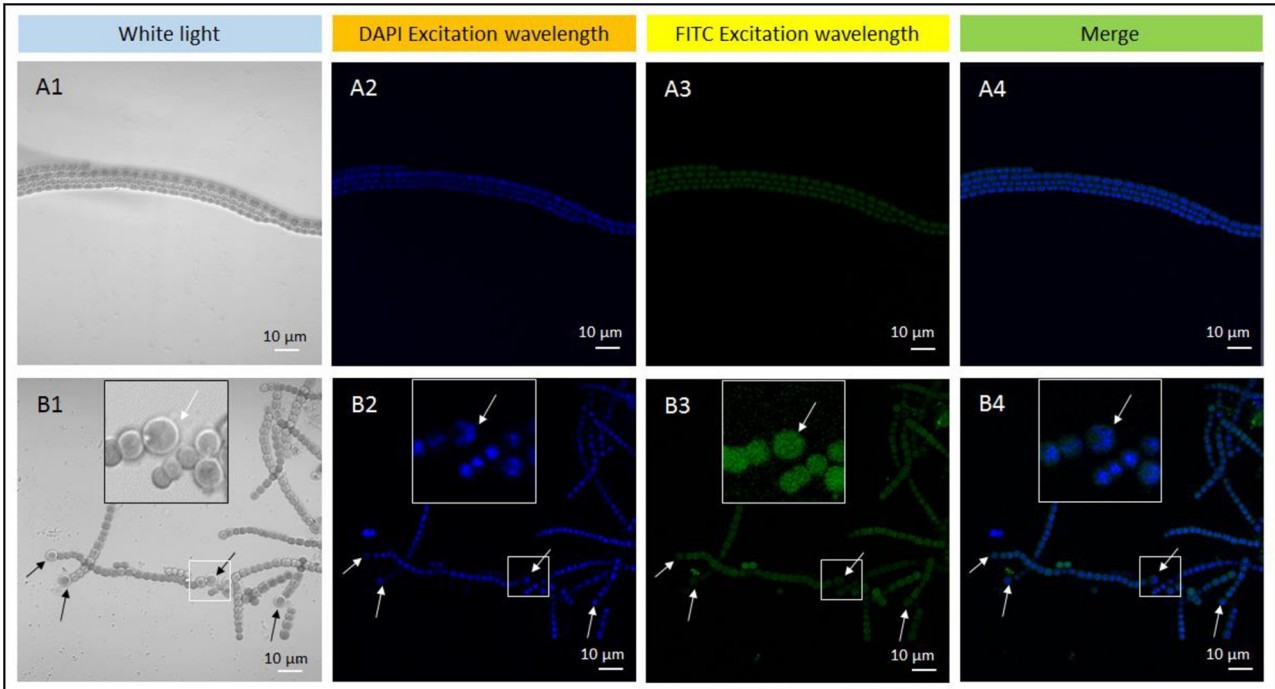

**Fig 7. Immunofluorescence analysis of nitrogenase iron protein (NifH) in filaments of *Anabaena* sp. strain PCC 7120 under nitrogen deprivation and nitrogen-replete condition.** (A1-A4) Filaments only consisting of vegetative cells under nitrogen-replete condition. A1, vegetative cells under white light; A2 and A3, vegetative cells was excited at 365 nm (DAPI excitation wavelength) and 488 nm (FITC excitation wavelength) respectively; A4, merged imagine of A2 and A3. (B1-B4) Filaments consisting of vegetative cells and heterocysts under nitrogen deprivation condition. B1, vegetative cells and heterocysts under white light; B2 and B3, vegetative cells and heterocysts were excited at 365 nm and 488 nm respectively; B4, merged imagine of B2 and B3. Double fluorescent labeling were used in the experiment. DAPI was used to visualize the DNA, and FITC was used to visualize NifH. DAPI, 4', 6-diamidino-2-phenylindole; FITC, fluorescein 5-isothiocyanate. In B1-B4, arrows show locations of heterocysts; the large square frame is an enlarged view of the small square frame.

maintain a suitable depth in an aqueous environment [24]. Under conditions of nitrogen-replete conditions, cyanobacterial filaments consisted of only vegetative cells (Fig 1A); under nitrogen deprivation conditions, these filaments consisted of both vegetative cells and heterocysts (Fig 1B). These results suggest that nitrogen deprivation induced heterocyst differentiation, which is consistent with previously reported findings [27]. In total, we identified eight DEGs encoding the gas vesicle protein (Table 2). In the comparison of NVs vs. NDVs, six

**Table 2. DEGs encoding gas vesicle proteins in three comparison pairs.**

| Gene ID | ORF ID | Log$_2$FC (NV vs. NDV) | Log$_2$FC (NV vs. NDH) | Log$_2$FC (NDV vs. NDH) | Length bp | Gene annotation |
|---------|--------|------------------------|------------------------|-------------------------|-----------|-----------------|
| gene2239 | all2247 | -1.25* | -0.48 | 0.76 | 377 | *GvpG* |
| gene2240 | all2248 | -1.59* | -0.84 | 0.75 | 734 | *GvpF* |
| gene2241 | all2249 | -3.15** | -1.76* | 1.38* | 473 | *GvpK* |
| gene2242 | all2251 | -3.05** | -1.30* | 1.74* | 446 | *GvpN* |
| gene2243 | all2250 | -1.67* | -1.76* | -0.10 | 1232 | *GvpJ* |
| gene2244 | all2252 | -3.87** | -1.00* | 2.86* | 389 | *GvpC* |
| gene2245 | asl2253 | -3.52** | -0.54 | - | 215 | *GvpB* |
| gene2246 | asl2254 | -3.86** | -0.78 | - | 215 | *GvpA* |

Note: Log$_2$FC, Log$_2$fold-change.

*, differently expressed gene with $1 < |Log_2FC| < 3$;

**, highly differently expressed gene with $Log_2FC \geq 3$. *Gvp*, gas vesicle synthesis protein gene. In three paired comparisons, X vs. Y refers to Y/X.

DEGs were highly downregulated, suggesting that nitrogen deprivation inhibits the expression of gas vesicle genes in vegetative cells; these results are consistent with most of the results obtained previously on gas vesicle protein regulation [28]. Furthermore, in the NDV vs. NDH comparison, three DEGs encoding gas vesicle genes were upregulated, suggesting that the expression of gas vesicle genes is higher in heterocysts than in vegetative cells under nitrogen deprivation conditions (Table 2). Thus, the density of vegetative cells increases via inhibition of gas vesicle protein expression under nitrogen deprivation conditions. Our results are consistent with the phenomenon of water bloom under eutrophication conditions [29, 30] and improve the understanding of the formation mechanism of water blooms at the transcription level.

## Heterocysts may perform photosynthetic functions under nitrogen deprivation conditions

In heterocyst-forming cyanobacteria, thylakoid membranes in vegetative cells contain photosystem I (PS-I) and PS-II, which use water as the electron donor and produce oxygen. Whether heterocysts have both PS-I and PS-II remains controversial [31]. Thylakoid membranes in vegetative cells have been suggested to contain PS-I and PS-II, whereas those in heterocysts predominantly contain PS-I [32]. The water-oxidizing activity of PS-II, as well as fluorescence emission at 685 nm associated with PS-II, has been reported to be absent in intact heterocysts from several filamentous strains [33, 34]. The reaction center proteins of PS-II were suggested to be degraded during heterocyst differentiation, although PS-II proteins can still be found in mature heterocysts [35–37]. In our study, KEGG enrichment analysis showed that carbon fixation in the photosynthetic pathway (ko00710) was significantly enriched in the NV vs. NDV comparison, suggesting that nitrogen deprivation alters the photosynthetic capability of *Anabaena* sp. strain PCC 7120 (Table 1). We also detected the expression of 13 and 19 genes encoding PS-I and PS-II proteins, respectively (S3 Table), including two DEGs encoding PS-I and five DEGs encoding PS-II. Comparison of NV vs. NDV suggested that nitrogen deprivation inhibited the expression of both genes encoding PS-I and PS-II, as shown by downregulation of *psaB*/*K* in PS-I and *psbN* in PS-II. Under nitrogen deprivation conditions, the expression of PS-I and PS-II genes was more active in heterocysts than in vegetative cells, shown by upregulation of *isiA* (PS-II), *psbAII* (PS-II), *psbAIII* (PS-II), *psbAIV* (PS-II), and *psbN* (PS-II). Thus, both PS-I- and PS-II-related genes were expressed in heterocysts, which is not consistent with the results of a previous study reporting that heterocysts lack PS-II [31–35].

## Nitrogen fixation capability of vegetative cells is promoted by nitrogen deprivation

Heterocysts differentiate from vegetative cells of some filamentous cyanobacteria to fix nitrogen for the entire filament under oxic growth conditions [38]. *Anabaena* sp. strain PCC 7120 contains two Mo-dependent nitrogenases [38, 39]. Among these, *nif1* is expressed exclusively in heterocysts and functions under oxic growth conditions [40], whereas *nif2* is expressed only under anoxic conditions in vegetative cells shortly after nitrogen step-down and long before heterocysts are formed [38]. Both nitrogenases fix nitrogen and supply it to the filament for growth [40]. We identified 23 nitrogen fixing genes, 21 of which were differentially expressed, including 10 highly-modulated DEGs (S4 Table). GO enrichment analysis of the DEGs showed that most of these genes encoded proteins with cofactor transferase activity, cofactor binding, and catalytic activity (Fig 4); these results are consistent with the properties of the nitrogenase-processing Mo cofactor. KEGG enrichment analysis indicated that multiple nitrogen-related

KEGG pathways were significantly enriched, including amino sugar and nucleotide sugar metabolism (ko00520), purine metabolism (ko00230), DNA replication (ko03030), and arginine biosynthesis (ko00220) (Table 1). Cluster analysis revealed that the DEGs were grouped into cluster 1, indicating higher gene expression in NDV and NDH samples than in the NV sample. Comparison of NVs vs. NDVs showed that *FKXN*, *NifD*, *NifE*, *NifH1*, *NifK*, *NifN*, and *NifX* were highly expressed in the NDV sample (S4 Table), suggesting that this set of DEGs mainly contributed to promoting nitrogen fixation under nitrogen deprivation conditions. Immunofluorescence analysis of NifH indicated that its expression in vegetative cells was low when the nitrogen supply is sufficient; moreover, nitrogen deprivation induced and promoted the expression of NifH in both vegetative cells and heterocysts (Fig 7). These results confirm the nitrogen fixation function of this set of DEGs in NDVs.

## Conclusion

Overall, our data confirm that directional RNA deep sequencing is a more thorough method of analyzing transcriptional regulation in heterocysts and vegetative cells by cell separation procedures, providing a systematic view of transcriptome level differences between heterocysts and vegetative cells under conditions of nitrogen step-down. Our results show that heterocysts and vegetative cells function by coordinating activities related to photosynthesis and nitrogen fixation. This direct comparison of transcript levels in NVs, NDVs, and NDHs revealed many characterized genes that are differentially regulated in the three cell types which should be further studied in heterocyst-forming cyanobacteria under nitrogen deprivation conditions.

## Materials and methods

### Experimental strain and treatments

*Anabaena* sp. strain PCC 7120 is a model strain useful for performing molecular studies of cell differentiation and heterocyst-forming cyanobacteria; this strain was purchased from the Freshwater Algae Culture Collection at the Institute of Hydrobiology with accession number of FACHB-418. The initial strain was cultured in Blue-Green medium (BG11) for 3 days, and then centrifuged and used in subsequent experiments. Nitrogen-replete medium (BG11) [41] and nitrogen deprivation medium (BG110) were used to culture PCC 7120 at an initial inoculation concentration $OD_{730}$ of 0.1. Before inoculation, the strain was centrifuged and washed three times with BG11 or BG110 for purification and enrichment. The culture period was 72 h. To maintain the relative stability of nutrient elements in the culture medium, the strains were centrifuged and replaced with fresh culture medium every 24 h. After 72 h of culture, the strain $OD_{730}$ in BG11 medium and BG110 medium were 0.997 and 0.621, respectively. The strain was cultured in 250-mL flasks sealed with plastic ventilating film, and the culture volume was 100 mL. PCC 7120 cultivation was carried out in a double layer shaping incubator under the following conditions: light of 300 $\mu mol \cdot m^{-2} \cdot s^{-1}$, 30°C, and shaking at a speed of 140 rpm. All centrifugation steps were performed at 500 ×$g$ for 5 min. After 72 h of cultivation, the formation of vegetative cells and heterocysts in the filaments was confirmed by laser scanning confocal microscopy (TCS SP2, Leica Microsystems, Wetzlar, Germany) (Fig 1A1, 1A2, 1B1 and 1B2).

### RNA extraction from vegetative cells and heterocysts

In total, three RNA samples (NDV, NDH, and NV) were used for extraction, each with three biological replicates. "NDV" refers to vegetative cells of PCC 7120 cultured in BG110 (N-deficient medium), "NDH" refers to heterocysts of the PCC 7120 strain cultured in BG110,

and "NV" refers to vegetative cells of the PCC 7120 strain cultured in BG11 (nitrogen-replete medium).

RNA was extracted from NDVs and NVs as described by Golden et al [42] with some modifications: 500 mL NDVs and NVs was sedimented at 500 ×*g* for 5 min at 4˚C. To ensure that only vegetative cells were broken down, the concentration of lysozyme was low and enzymolysis time was short. The precipitate was resuspended in 1 mL lysozyme solution (1 mg/mL lysozyme in TE buffer) at 30˚C for 5 min, shaking each tube for 15 s at 1-min intervals. Next, 5 mL of guanidine isothiocyanate cell lysis solution (mixed solution in the EASYspin Plus Bacteria RNA kit, Aidlab, Beijing, China) was added to terminate the reaction as quickly as possible. This solution was centrifuged to remove the precipitate, and the supernatants were used for RNA isolation according to the instructions of the EASYspin Plus Bacteria RNA kit (Aidlab). The sediment obtained after this treatment contained vegetative cells and heterocysts (Fig 1C1 and 1C2).

NDHs were purified and enriched using as described by Golden et al [42] with some modifications: 500 mL NDHs was cultured in BG110 for 72 h and sedimented at 500 ×g for 5 min at 4˚C. To eliminate contamination by vegetative cells, NDHs were treated with a high concentration of lysozyme, Triton X-100, and long-term cell lysis to ensure that all vegetative cells were completely lysed. The precipitate was re-suspended in 50 mL lysozyme solution for 1 h at 25˚C (5 mg/mL lysozyme and 0.1% Triton X-100 in TE buffer). Most vegetative cells were lysed during this period, whereas the heterocysts remained intact, as determined by laser scanning confocal microscopy (Leica TCS SP2) (Fig 1D1 and 1D2). The microscopy results showed that heterocysts were mixed with impurities produced by lysed vegetative cells because of the strong adhesion of heterocysts (Fig 1D1 and 1D2). This phenomena were also observed during heterocyst isolation by Park et al [22], who suggested that the impurities were ruptured remains of vegetative cell or heterocyst protoplasts, and heterocysts mixed with cell contents were used to extract RNA from the heterocysts. To eliminate possible RNA residue from the ruptured remains of vegetative cells, the heterocyst-containing sediment was re-suspended in 500 μL TE buffer containing 50 μg/mL RNAase at 37˚C for 1 h, followed by three washes with TE buffer. Finally, the purified heterocysts were mixed with mill silica sand (0.6–1.5 mm), ground into powder in liquid nitrogen, and diluted in deionized water for RNA extraction according to the instructions of the kit mentioned previously.

## RNA extraction, library construction, and sequencing

To investigate the changes in gene expression in NDVs, NDHs, and NVs, nine DGE profiling libraries were constructed using RNA samples of NDV1, NDV2, NDV3, NDH1, NDH2, NDH3, NV1, NV2, and NV3. RNA pretreatment was performed before sequencing, including rRNA depletion, RNA fragmentation, cDNA synthesis, end repair, A-tailing, adapter ligation, and PCR (S1 File). rRNA depletion was performed using the Ribo-Zero Magnetic Kit for bacteria (Epicentre Biotechnologies, Madison, WI, USA). The samples were then cleaned using RNAclean XP beads (Beckman Coulter, Brea, CA, USA). The RNA was fragmented into 130–170 nt by adding fragmentation buffer (Ambion, Austin, TX, USA) to the samples and incubating the samples at 70˚C. The samples were purified again with RNAclean XP beads. Purified RNA was used for cDNA synthesis. First-strand cDNA was synthesized using First Strand Master Mix and Super Script II reverse transcriptase (Invitrogen, Carlsbad, CA, USA). The mixture was incubated at 42˚C for 50 min followed by inactivation at 70˚C for 15 min. Using second-strand Master mix, second-strand cDNA was synthesized. Before PCR, end repair and poly-A tail addition was performed using End Pair Repair Mix and A-tailing mix simultaneously. RNA index adapters were added to the adenylated 3′ end of the DNA using T4 DNA

ligase. This DNA sample was used for PCR amplification, which was performed using PCR Master Mix and a PCR primer cocktail, for several rounds. The library was validated using an Agilent 2100 Bioanalyzer (Agilent Technologies, Santa Clara, CA, USA) to determine the average molecular length. The profiling libraries were sequenced on a HiSeq 4000 system (Illumina, San Diego, CA, USA) at Shanghai Personal Biotechnology Co., Ltd. (Shanghai, China). The raw transcriptome data were deposited into the NCBI sequence read archive under accession number SAMN12877542-12877550.

## Data analysis and mapping of DGE tags

After filtering reads with an adaptor at the 3′ end and quality values less than Q20 using Cutadapt [43], clean reads were assembled. Functional annotation was performed using five databases including Non-Redundant Protein Sequence Database (NR), GO, KEGG, and evolutionary genealogy of genes: Non-supervised Orthologous Groups (eggnog) databases [44]. Three pairs of DEG profiles for different sample libraries (NV vs. NDH, NV vs. NDV, and NDV vs. NDH; in each paired comparison, the former was used as a control and the latter as the experimental group) were compared to identify DEGs in vegetative cells and heterocysts of *Anabaena* sp. strain PCC 7120.

After assembly, the clean reads were mapped to the reference *Nostoc* sp. PCC 7120 genome (https://www.ncbi.nlm.nih.gov/genome/genomes/13531. BioSample = SAMD00061094, BioProject = PRJNA244; BioSample = SAMN10102199, BioProject = PRJNA492407) and gene expression levels were calculated for each sample. DEGs were identified using the DESeq R package (version 1.18.0) using thresholds of $\log_2|\text{fold-change}| > 1$ and *P*-value $< 0.05$. KEGG pathway enrichment analysis of the DEGs was performed using BLAST by searching the KEGG database (http://www.kegg.jp/kegg/). $P \leq 0.05$ was set as the threshold for significant enrichment of KEGG pathways. GO functional enrichment of DEGs was performed using the topGO R package [45], and terms with a *P*-value less than 0.05 were regarded as significant. Hierarchical clustering analysis of the DEGs was performed using Multi Experiment Viewer (http://mev.tm4.org/#/welcome).

## qRT-PCR analysis

DEGs obtained by Illumina sequencing were verified by qRT-PCR. The RNA samples used for qRT-PCR were identical to those used for the DEG experiments. cDNA was synthesized using the TUREscript One Step RT-PCR Kit (Aidlab) according to the manufacturer's instructions. Primers were designed using Premier 6.0 software (Premier Biosoft, Palo Alto, CA, USA) and synthesized commercially (Sangon Biotech Shanghai Co., Ltd., Shanghai, China) (S5 Table). The 20-μL reaction mixture contained 1 μL cDNA template, 1 μL each primer (10 μM), and 10 μL 2× SYBR Green qPCR (Aidlab); qRT-PCR was performed using the 2× SYBR Green qPCR kit (Aidlab) according to the manufacturer's protocol, and relative expression levels of the genes were calculated by the $2^{-\Delta\Delta CT}$ method [46]. In this calculation, the relative mRNA levels of target genes were normalized by *rnpB* [47] using the ΔCT method [ΔCT = av CT(target gene)–av CT (*rnpB*)]. To determine the fold-change, gene expression in NVs was used as a control, and the results were calculated using the ΔΔCT (comparative threshold cycle) method: ΔΔCT = (av CT (target gene)–av CT(*rnpB*))sample–(av CT(target gene)–av CT(*rnpB*))control. In three paired comparisons, X vs. Y refers to Y/X. Y was used as the target gene and X was used as a control.

## Immunofluorescence analysis of nitrogenase iron protein

Immunofluorescence analysis of nitrogenase iron protein was performed as previously described with minor modifications [48]. After 72 h culture under nitrogen-replete and

deprivation conditions, strain PCC 7120 was fixed with 4% paraformaldehyde in phosphate-buffered saline (PBS) for 30 min at 25˚C. The samples were rinsed three times with PBS containing 0.1% Triton X-100 (PBST), and then incubated for 5 min in PBST at 25˚C. The samples were incubated for 1 h with 6 μg/mL affinity-purified anti-NifH antibody (Agrisera, Vaesterbotten, Sweden) diluted in PBST containing 1 mg/mL bovine serum albumin and three washes with PBST. The samples were incubated in the dark with rabbit anti-chicken IgY conjugated to FITC (Sigma, St. Louis, MO, USA) for 45 min and then washed twice with PBST. The samples were mounted in Vectashield medium containing DAPI (Vector Laboratories, Burlingame, CA, USA H-1500) and viewed under a laser scanning confocal microscope (Leica TCS SP2).

## Supporting information

**S1 File.**
(DOC)

**S1 Fig.** *NifH1* and *nifH2* gene expression analysis by RT-PCR.
(DOC)

**S1 Table. RNA-seq reads of *Anabaena* sp. strain PCC 7120 mapped to reference genome.**
(DOCX)

**S2 Table. Distribution statistics results of RNA-Seq mapped events for all reads.**
(DOCX)

**S3 Table. The DEGs encoding PS I, PS II in three comparison pairs.**
(DOCX)

**S4 Table. The DEGs encoding nitrogenase genes in three comparison pairs.**
(DOCX)

**S5 Table. Primers used for qRT-qPCR.**
(DOCX)

## Author Contributions

**Data curation:** Hongli He, Runyu Miao, Lilong Huang, Hongshan Jiang.

**Formal analysis:** Lilong Huang, Hongshan Jiang.

**Funding acquisition:** Yunqing Cheng.

**Methodology:** Hongli He, Runyu Miao, Lilong Huang, Hongshan Jiang, Yunqing Cheng.

**Writing – original draft:** Runyu Miao, Lilong Huang, Yunqing Cheng.

**Writing – review & editing:** Hongli He, Hongshan Jiang, Yunqing Cheng.

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
