## [Decision Letter · Decision Letter 0]

27 Nov 2020

PONE-D-20-27050

Genome-wide differential expression analysis of vegetative cell and heterocyst responses to nitrogen deprivation in the Anabaena sp. strain PCC 7120

PLOS ONE

Dear Dr. Cheng,

Thank you for submitting your manuscript to PLOS ONE. After careful consideration, we feel that it has merit but does not fully meet PLOS ONE’s publication criteria as it currently stands. Therefore, we invite you to submit a revised version of the manuscript that addresses the points raised during the review process.

Both the reviewers have suggested some additional experiments to improve the manuscript. I would suggest to submit the revised manuscript with data on suggested experiments. 

We look forward to receiving your revised manuscript.

Kind regards,

Anil Kumar Singh, Ph.D.

Academic Editor

PLOS ONE

Journal Requirements:

2.PLOS requires an ORCID iD for the corresponding author in Editorial Manager on papers submitted after December 6th, 2016. Please ensure that you have an ORCID iD and that it is validated in Editorial Manager. To do this, go to ‘Update my Information’ (in the upper left-hand corner of the main menu), and click on the Fetch/Validate link next to the ORCID field. This will take you to the ORCID site and allow you to create a new iD or authenticate a pre-existing iD in Editorial Manager. Please see the following video for instructions on linking an ORCID iD to your Editorial Manager account: https://www.youtube.com/watch?v=_xcclfuvtxQ

3. We note you have included a table to which you do not refer in the text of your manuscript. Please ensure that you refer to Table 5 in your text; if accepted, production will need this reference to link the reader to the Table.

<h1>** **</h1>

Reviewers' comments:

Reviewer's Responses to Questions

**Comments to the Author**

1. Is the manuscript technically sound, and do the data support the conclusions?

Reviewer #1: Yes

Reviewer #2: Partly

2. Has the statistical analysis been performed appropriately and rigorously? 

Reviewer #1: Yes

Reviewer #2: N/A

3. Have the authors made all data underlying the findings in their manuscript fully available?

Reviewer #1: Yes

Reviewer #2: Yes

4. Is the manuscript presented in an intelligible fashion and written in standard English?

Reviewer #1: No

Reviewer #2: Yes

5. Review Comments to the Author

Reviewer #1: The manuscript entitled " Genome-wide differential expression analysis of vegetative cell and heterocyst responses to nitrogen deprivation in the Anabaena sp. strain PCC 7120" is quite interesting and has a potential use of great knowledge to know the nitrogen fixing capabilities of Anabaena’s heterocyst and vegetative cell. Though the manuscript is only based on transcriptome level analysis, as per my consideration, the current manuscript can be considered in Plos one, it satisfies all the comments. Thus, I am suggesting a major revision.

Overall feedback:

1. Apart from the suggested comments can authors perform acetylene reduction assay or similar kind of experiment with purified NDV, NDH, and NV?? Nitrogenase functional analysis line ARA will make the manuscript much more applying and can support their preexisting experiment.

2. Overall language and sentence construction of the manuscript is very poor. Some of the compound sentences are very difficult to understand and changes the meaning of the paragraphs.

3. Good quality figures should be provided

Comments:

1. ABSTRACT: line no. 16. Reframe the sentences. First two sentences staring with in sounds odd.

2. line no. Reframe the sentences 53 to 56

3. line no. 56-59. Use simple sentence and avoid complex sentences.

4. line no. 77-79. The cited statement “not the only cells capable of nitrogen fixation in heterocystous cyanobacteria and that vegetative cells can fix molecular nitrogen at the same rate as heterocysts” is from a 50 years old paper!!! Authors are highly encouraged to go through recent studies based on modern techniques. In that paper (due to lack of technologies in those days), its not clear that wherever its fixed nitrogen or the transported nitrogen form the heterocysts entered the vegetative cell!!!!

5. Line 92. Use the word strain or full name of PCC 7120.

6. Line 275. Dose the culture have any accession number/ registration number from Institute of Hydrobiology, Chinese Academy of Sciences? Please provide it in the manuscript.

7. Incubation temperature is missing in Experimental Strain and Treatments section.

8. I think separation of NDV and NDH should be added before RNA Extraction to make it more meaningful. (L290)

9. Can authors include Bioanalyzer reading in supplementary materials?? Its very important to reproduce these kind of experiment aging and again.

10. L 354 Nostoc will be italics

11. For qRT-PCR analysis, how the values were normalized?

12. Table 1,2, 5,6 and 7 should be in supplementary material or can be presented in more meaningful way.

13. Quality of all the figures are really poor and it’s not publishable. In fig. 4,5,6 I can’t read anything. Hight quality figures should be provided.

Suggestion:

Title should be modified by stating the main finding of the study

Reviewer #2: The manuscript entitled “Genome-wide differential expression analysis of vegetative cell and heterocyst responses to nitrogen deprivation in the Anabaena sp. strain PCC 7120” is interesting, however it has plenty of flaws, out of which I am pointing few below.

Technical comments

1. It is well known fact that in Anabaena, N-fixation occurred in heterocyst whereas authors tried to prove N-fixing ability in vegetative cells of Anabaena sp. strain PCC 7120 by using transcriptome analysis, however, a simple test i.e. Acetylene Reduction Assay (ARA) and nifH-PCR could be a better option for preliminary screening which are missing in this study. Please provide these data.

2. Authors also did not prove how N-fixation occurs in vegetative cells of Anabaena sp. strain PCC 7120.

3. Whether heterocysts have both PS-I and PS-II remains controversial…justify the statements with references.

4. If PSII expressed in heterocyst then what is mechanism to prevent nitrogenase oxidation inside heterocyst.

6. PLOS authors have the option to publish the peer review history of their article (what does this mean?). If published, this will include your full peer review and any attached files.

Reviewer #1: **Yes: **Dr. Avishek Banik

Reviewer #2: No

---

## [Author Response · Author response to Decision Letter 0]

31 Jan 2021

We would like to express our sincere thanks to the reviewers for their time and constructive suggesting. For your guidance, itemized response to reviewer’ comments is appended below. 

Response to the reviewer 1:

General Comments: The manuscript entitled "Genome-wide differential expression analysis of vegetative cell and heterocyst responses to nitrogen deprivation in the Anabaena sp. strain PCC 7120" is quite interesting and has a potential use of great knowledge to know the nitrogen fixing capabilities of Anabaena’s heterocyst and vegetative cell. Though the manuscript is only based on transcriptome level analysis, as per my consideration, the current manuscript can be considered in Plos one, it satisfies all the comments. Thus, I am suggesting a major revision.

Response:

We appreciate the reviewer’s positive evaluation of our work. We have revised the manuscript to incorporate your suggestions and comments. The method has been better explained and each step has been given with more detail. The tissue to which the method was applied has been described more accurately. We have improved the English writing to reduce the grammar mistakes and inaccurate description as far as possible. The data was added to make it easier to read and conclusions easier to see.

Overall feedback:

 1. Apart from the suggested comments can authors perform acetylene reduction assay or similar kind of experiment with purified NDV, NDH, and NV?? Nitrogenase functional analysis line ARA will make the manuscript much more applying and can support their preexisting experiment.

Thank you for suggestion. Nitrogenase activity was measured as acetylene reduction assay (ARA) of the cell samples in culture tubes containing 2mL BG 11 or BG110 medium capped with subha seals. The cell samples used for ARA were NDVs+NDHs in BG110, NDHs in BG110, and NVs in BG11. Acetylene at 10% of gas phase was injected followed by incubation under fluorescent light for 4 h. 50 µL gas phase was measured with GC-2010 Shimadzu gas chromatograph (GC-2010, Shimadzu Corporation, Japan) equipped with a DB-WAX capillary column (30 m×0.25 mm, 0.25 μm), carrier gas nitrogen 20 mL. min–1, column temperature 100ºC, injector temperature 130ºC, detector temperature 230ºC). Nitrogenase activity was expressed as nmol.2mL culture.（4h）–1. The direct output of GC is as follows: 

However, we can’t obtain pure NDV cells for the nitrogenase activity analysis because the technique for the isolation of heterocysts of Anabaena cylindrica was all based on vegetable cell disruption. We didn't get the expected result about nitrogenase activity analysis, the pure NDH cells showed lower nitrogenase activity compared with the NDV+NDH (The whole filaments in BG110). We think that the nitrogenase ability of pure NDH cells should be different with NDH in filaments. Therefore, the ARA results of different cells only showed in the response letter.

2. Overall language and sentence construction of the manuscript is very poor. Some of the compound sentences are very difficult to understand and changes the meaning of the paragraphs.

Thank you for your careful review. We are very sorry for the poor language and sentence in this manuscript and inconvenience they caused in your reading. The compound sentences has been thoroughly revised and rewritten by a native English editor from Editage Company, so we hope it can meet the journal’s standard. 

3. Good quality figures should be provided

Accepted; the poor figure 4, 5 and 6 has been replaced in the revised paper.

1. ABSTRACT: line no. 16. Reframe the sentences. First two sentences staring with in sounds odd.

Accepted, the first two sentences you mentioned have been deleted.

2. line no. Reframe the sentences 53 to 56 

The sentences you mentioned haven been changed. “Cyanobacteria evolved approximately 3.5 billion years ago. As the first photosynthetic prokaryotes on earth, cyanobacteria can efficiently fix atmospheric nitrogen via a process catalyzed by nitrogenase [1]. Nitrogenase-catalyzed substrate reduction requires the association of an iron (Fe) protein and molybdenum-iron (MoFe) protein, with electron transfer occurring from the Fe protein to the MoFe protein. However, nitrogenase responsible for the reduction of N2 is extremely sensitive to O2 [2-5].” See line 39-44

3. line no. 56-59. Use simple sentence and avoid complex sentences.

Accepted. We have made Corresponding changes. “How to avoid oxygen released during photosynthesis by nitrogenase has attracted much attention.”See line 44-45

4. line no. 77-79. The cited statement “not the only cells capable of nitrogen fixation in heterocystous cyanobacteria and that vegetative cells can fix molecular nitrogen at the same rate as heterocysts” is from a 50 years old paper!!! Authors are highly encouraged to go through recent studies based on modern techniques. In that paper (due to lack of technologies in those days), its not clear that wherever its fixed nitrogen or the transported nitrogen form the heterocysts entered the vegetative cell!!!!

A relative new reference has been cited according to your suggestion. 

Thiel T, Lyons E M, Erker J C, Ernst A. 1995. A second nitrogenase in vegetative cells of a heterocyst-forming cyanobacterium. Proc Natl Acad Sci USA 92:9358-9362.

5. Line 92. Use the word strain or full name of PCC 7120.

Accepted. See line 85.

6. Line 275. Dose the culture have any accession number/ registration number from Institute of Hydrobiology, Chinese Academy of Sciences? Please provide it in the manuscript.

It was purchased from Freshwater Algae Culture Collection at the Institute of Hydrobiology with accession number of FACHB-418. See line 251-254.

http://algae.ihb.ac.cn/english/algaeDetail.aspx?id=258

7. Incubation temperature is missing in Experimental Strain and Treatments section.

Accepted. Incubation temperature is 30 ℃. See line 263.

8. I think separation of NDV and NDH should be added before RNA Extraction to make it more meaningful. (L290)

Knowledge of the nature of the heterocysts of blue-green algae has been limited by the lack of suitable techniques for investigation of the physiology and biochemistry of these puzzling structures. Some successful technique for the isolation of heterocysts of Anabaena cylindrica was all based on ‘differential cell disruption’. Therefore, it is difficult for separation of NDV and NDH to acquire the intact vegetative cells.

9. Can authors include Bioanalyzer reading in supplementary materials?? Its very important to reproduce these kind of experiment aging and again.

We are puzzled by the expression of "Bioanalyzer reading". In order to let other researchers repeat this experiment, we think that the reviewers mentioned the parameters used in transcriptome analysis, specifically, the parameters of software used in transcriptome analysis. To address this question, software and parameters used in our study have include a supplementary material (supplementary file 1) in the revised manuscript.

10. L354 Nostoc will be italics.

Accepted. All the “Nostoc italics” have been italicized. 

11. For qRT-PCR analysis, how the values were normalized?

In the process of calculation, relative mRNA levels of target genes were normalized by rnpB (46) using the ΔCT method [ΔCT = av CT(target gene) – av CT (rnpB)]. To determine the fold-change, the gene expression in NVs was used as control, results were calculated using the ΔΔCT (comparative threshold cycle) method, ΔΔCT = (av CT(target gene) – av CT(rnpB))sample – (av CT(target gene) – av CT(rnpB))control. In three paired comparisons, X vs. Y refers to Y/X. Y was used as target gene and the X was used as control. The process of calculation has added in revised manuscript (See line 349-354)

12. Table 1,2, 5,6 and 7 should be in supplementary material or can be presented in more meaningful way.

Accepted. Table 1,2, 5,6 and 7 have been put in supplementary materials as you suggested.(see table S1- table S5)

13. Quality of all the figures are really poor and it’s not publishable. In fig. 4, 5, 6 I can’t read anything. Hight quality figures should be provided.

 � Accepted. Fig. 4, 5, 6 has been replaced with high quality figures. 

Suggestion:

Title should be modified by stating the main finding of the study

Accepted. Vegetative cells may perform nitrogen fixation function under nitrogen deprivation in Anabaena sp. strain PCC 7120 based on genome-wide differential expression analysis.

#2: The manuscript entitled “Genome-wide differential expression analysis of vegetative cell and heterocyst in the Anabaena sp. strain PCC 7120” is interesting, however it has plenty of flaws, out of which I am pointing few below.

We appreciate the reviewer’s positive evaluation of our work.

Technical comments

1. It is well known fact that in Anabaena, N-fixation occurred in heterocyst whereas authors tried to prove N-fixing ability in vegetative cells of Anabaena sp. strain PCC 7120 by using transcriptome analysis, however, a simple test i.e. Acetylene Reduction Assay (ARA) and nifH-PCR could be a better option for preliminary screening which are missing in this study. Please provide these data.

Nitrogenase activity was measured as acetylene reduction assay (ARA) of the cell samples in culture tubes containing 2mL BG 11 or BG110 medium capped with subha seals. The cell samples used for ARA were NDVs+NDHs in BG110, NDHs in BG110, and NVs in BG11. Acetylene at 10% of gas phase was injected followed by incubation under fluorescent light for 4 h. 50 µL gas phase was measured with GC-2010 Shimadzu gas chromatograph (GC-2010, Shimadzu Corporation, Japan) equipped with a DB-WAX capillary column (30 m×0.25 mm, 0.25 μm), carrier gas nitrogen 20 mL. min–1, column temperature 100ºC, injector temperature 130ºC, detector temperature 230ºC). Nitrogenase activity was expressed as nmol.2mL culture.（4h）–1. The direct output of GC is as follows: 

However, we can’t obtain pure NDV cells for the nitrogenase activity analysis because the technique for the isolation of heterocysts of Anabaena cylindrica was all based on vegetable cell disruption. We didn't get the expected result about nitrogenase activity analysis, the pure NDH cells showed lower nitrogenase activity compared with the NDV+NDH (The whole filaments in BG110). We think that the nitrogenase ability of pure NDH cells should be different with NDH in filaments. Therefore, the ARA results of different cells only showed in the response letter.

We think that PCR is not a good method to prove the differential expression of Nif gene. No matter whether the gene is expressed or not, the Nif gene in the genome will be amplified by PCR. We speculated that the reviewer wanted to supplement the results of qRT-PCR or RT-PCR. qRT-PCR is shown in Figure 6 of the paper. Here, we added some electrophoresis pictures of RT-PCR, which proved that the expression trend of NifH1 and NifH2 genes in vegetative cells and heterospores was similar to that of transcriptome and qRT-PCR. See figure S1.

2. Authors also did not prove how N-fixation occurs in vegetative cells of Anabaena sp. strain PCC 7120.

As a result of our study, we reached the conclusion that N-fixation occurs in vegetative cells of Anabaena sp. strain PCC 7120 by comparative transcriptome analysis and histological analysis. Knowledge of the nature of the heterocysts of blue-green algae has been limited by the lack of suitable techniques for investigation of the physiology and biochemistry of these puzzling structures. Some successful technique for the isolation of heterocysts of Anabaena cylindrica was all based on ‘differential cell disruption’. Therefore it is difficult for us and all related researchers to separation of NDV and NDH to acquire the intact vegetative cells to study the N-fixation function.

3. Whether heterocysts have both PS-I and PS-II remains controversial…justify the statements with references.

The reference has been added in revised paper as the 31th reference. Ferimazova et al confirmed mature heterocysts were found to have an intact PSII in vivo at single-cell level by chlorophyll fluorescence kinetic microscopy, and the possible importance of the functional PS II in heterocysts is discussed [Ferimazova et al. Photosynthesis Research, 2013, 116(1): 79-91. https://doi.org/10.1007/s11120-013-9897-z]. We conclude that both PS-I- and PS-II-related genes were expressed in heterospores, which is inconsistent with a previous study that reported that heterocysts lacked PS-II (Cardona et al, 2009, Tel-Or and Stewart, 1977, Thomas, 1972). Conclusively, we can say that at least PS-II inactivation or degraded did not occur at the transcription level.

4. If PSII expressed in heterocyst then what is mechanism to prevent nitrogenase oxidation inside heterocyst.

Nitrogenase oxidation is an interesting and important question, and we are also puzzled by the question. In the nonheterocystous cyanobacteria, the O2 evolved during photosynthesis and nitrogenase was confirmed to discrete in subcellular bodies identified as cyanophycin granules (Finzi-Hart et al., PNAS April 14, 2009,106 (15) 6345-6350; https://doi.org/10.1073/pnas.0810547106). In the heterocystous cyanobacteria, we believe that they have their similar or unique oxygen suppression mechanism under nitrogen deficiency. On the other hand, we observed that the filaments tend to float in liquid medium when nitrogen supply is sufficient (Figure A); in contrast, most of the filaments tend to sink in the culture solution, whereas some small particles were found to be suspended (Figure B). Thus, the increase of whole filaments density (which sink in the culture solution) should be benefit to prevent nitrogenase oxidation under the conditions of nitrogen deprivation. Our study results are consistent with the phenomenon of water bloom under eutrophication (Ekvall et al, 2013). Because oxygen suppression mechanism under nitrogen deficiency is only our own conjecture, and there is no systematic and credible experimental evidence, we did not write the above conjecture in the discussion.

---

## [Decision Letter · Decision Letter 1]

22 Feb 2021

Vegetative cells may perform nitrogen fixation function under nitrogen deprivation in Anabaena sp. strain PCC 7120 based on g enome-wide differential expression analysis

PONE-D-20-27050R1

Dear Dr. Cheng,

We’re pleased to inform you that your manuscript has been judged scientifically suitable for publication and will be formally accepted for publication once it meets all outstanding technical requirements.

Kind regards,

Anil Kumar Singh, Ph.D.

Academic Editor

PLOS ONE

Additional Editor Comments (optional):

Reviewers' comments:

Reviewer's Responses to Questions

**Comments to the Author**

1. If the authors have adequately addressed your comments raised in a previous round of review and you feel that this manuscript is now acceptable for publication, you may indicate that here to bypass the “Comments to the Author” section, enter your conflict of interest statement in the “Confidential to Editor” section, and submit your "Accept" recommendation.

Reviewer #1: All comments have been addressed

Reviewer #2: All comments have been addressed

2. Is the manuscript technically sound, and do the data support the conclusions?

Reviewer #1: Yes

Reviewer #2: (No Response)

3. Has the statistical analysis been performed appropriately and rigorously? 

Reviewer #1: Yes

Reviewer #2: (No Response)

4. Have the authors made all data underlying the findings in their manuscript fully available?

Reviewer #1: Yes

Reviewer #2: (No Response)

5. Is the manuscript presented in an intelligible fashion and written in standard English?

Reviewer #1: Yes

Reviewer #2: (No Response)

6. Review Comments to the Author

Reviewer #1: All most all the issues raised by the reviewers are well addressed. Though authors put several references in their defense but as per my consideration, several other experiment should be set up to confer the PSII of heterocyst and it mechanism to prevent nitrogenase oxidation inside. The positive ARA activity is an indication that the Vegetative cells may perform nitrogen fixation under limited nitrogen condition. Hope this paper will provide a new direction of research to the authors as well as the researchers who will read it in near future.

Best

Reviewer #2: (No Response)

7. PLOS authors have the option to publish the peer review history of their article (what does this mean?). If published, this will include your full peer review and any attached files.

Reviewer #1: **Yes: **Dr. Avishek Banik

Reviewer #2: **Yes: **Dr. Upendra Kumar

---

## [Editor Report · Acceptance letter]

25 Feb 2021

PONE-D-20-27050R1 

Vegetative cells may perform nitrogen fixation function under nitrogen deprivation in *Anabaena* sp. strain PCC 7120 based on genome-wide differential expression analysis 

Dear Dr. Cheng:

I'm pleased to inform you that your manuscript has been deemed suitable for publication in PLOS ONE. Congratulations! Your manuscript is now with our production department. 

Kind regards, 

on behalf of

Dr. Anil Kumar Singh 

Academic Editor

PLOS ONE